# Impact of Histotripsy on Development of Intrahepatic Metastases in a Rodent Liver Tumor Model

**DOI:** 10.3390/cancers14071612

**Published:** 2022-03-22

**Authors:** Tejaswi Worlikar, Man Zhang, Anutosh Ganguly, Timothy L. Hall, Jiaqi Shi, Lili Zhao, Fred T. Lee, Mishal Mendiratta-Lala, Clifford S. Cho, Zhen Xu

**Affiliations:** 1Department of Biomedical Engineering, University of Michigan, Ann Arbor, MI 48109, USA; wtejaswi@umich.edu (T.W.); hallt@umich.edu (T.L.H.); 2Department of Radiology, University of Michigan, Ann Arbor, MI 48109, USA; maggiez@med.umich.edu (M.Z.); mmendira@med.umich.edu (M.M.-L.); 3Department of Surgery, University of Michigan, Ann Arbor, MI 48109, USA; ganutosh@med.umich.edu (A.G.); cliffcho@med.umich.edu (C.S.C.); 4Department of Pathology & Clinical Labs, Rogel Cancer Center, University of Michigan, Ann Arbor, MI 48109, USA; jiaqis@med.umich.edu; 5Department of Biostatistics, University of Michigan, Ann Arbor, MI 48109, USA; zhaolili@med.umich.edu; 6Department of Radiology, University of Wisconsin, Madison, WI 53705, USA; flee@uwhealth.org; 7Department of Surgery, Ann Arbor VA Healthcare, Ann Arbor, MI 48105, USA

**Keywords:** histotripsy, hepatocellular carcinoma, tumor ablation, immunomodulation, metastases

## Abstract

**Simple Summary:**

Histotripsy is a novel technique that mechanically disrupts tumors, through precisely controlled acoustic cavitation. There is insufficient evidence regarding the effects of histotripsy on the risk of recurrence and metastases, following tumor debulking. The aim of this study is to evaluate the effect of partial histotripsy tumor ablation (~50–75% tumor volume targeted) on untargeted tumor progression, survival outcomes, risk of metastases and immune infiltration, in an orthotopic, immunocompetent, metastatic rodent hepatocellular carcinoma (HCC) model. Even with partial ablation, complete local tumor regression was observed in 81% of treatment rats, with no recurrence or metastasis. In contrast, 100% of the untreated control animals showed local tumor progression and intrahepatic metastases. Histotripsy-treated animals had statistically significant improved survival outcomes compared to controls (*p*-value < 0.0001). Histotripsy-treated animals had increased immune infiltration, as compared to controls, which may have contributed to the eventual regression of the untargeted tumor region in partial histotripsy-treated tumors.

**Abstract:**

Histotripsy has been used for tumor ablation, through controlled, non-invasive acoustic cavitation. This is the first study to evaluate the impact of partial histotripsy ablation on immune infiltration, survival outcomes, and metastasis development, in an in vivo orthotopic, immunocompetent rat HCC model (McA-RH7777). At 7–9 days post-tumor inoculation, the tumor grew to 5–10 mm, and ~50–75% tumor volume was treated by ultrasound-guided histotripsy, by delivering 1–2 cycle histotripsy pulses at 100 Hz PRF (focal peak negative pressure P– >30 MPa), using a custom 1 MHz transducer. Complete local tumor regression was observed on MRI in 9/11 histotripsy-treated rats, with no local recurrence or metastasis up to the 12-week study end point, and only a <1 mm residual scar tissue observed on histology. In comparison, 100% of untreated control animals demonstrated local tumor progression, developed intrahepatic metastases, and were euthanized at 1–3 weeks. Survival outcomes in histotripsy-treated animals were significantly improved compared to controls (*p*-value < 0.0001). There was evidence of potentially epithelial-to-mesenchymal transition (EMT) in control tumor and tissue healing in histotripsy-treated tumors. At 2- and 7-days post-histotripsy, increased immune infiltration of CD11b^+^, CD8^+^ and NK cells was observed, as compared to controls, which may have contributed to the eventual regression of the untargeted tumor region in histotripsy-treated tumors.

## 1. Introduction

Liver cancer is one of the top ten causes of cancer-related deaths worldwide and in the United States [1]. Hepatocellular carcinoma (HCC) accounts for 75% of all liver cancer cases, most frequently occurring in patients with chronic liver diseases, from etiologies such as hepatitis B and C, alcohol abuse, non-alcoholic steatohepatitis (NASH) and nonalcoholic fatty liver disease (NAFLD), resulting in cirrhosis [2]. The liver is also a frequent site for metastases originating from colorectal cancer, pancreatic cancer, melanoma, lung cancer and breast cancer [3]. Depending on the location, severity and staging of liver cancer, multiple treatment options are currently available, including surgical resection, liver transplantation, ablation techniques (including radiofrequency ablation (RFA), microwave ablation (MWA), high-intensity focused ultrasound (HIFU), cryoablation), chemotherapy, radiation therapy, targeted drug therapy and immunotherapies [4]. Even with current medical treatments, the 5-year patient survival rate in the United States is only 20%, the second lowest amongst all cancers [1]. Symptoms associated with liver cancer may not show at early stages, placing the patients at an increased risk for nodal and distant metastases, which further lowers their 5-year survival rate to an estimated 3–11% [5]. Even after treatment, high prevalence of tumor recurrence and metastasis [6] highlights the clinical need for improving outcomes of liver cancer. In fact, metastasis accounts for >90% of all cancer-associated deaths, and metastatic progression is predominantly regulated by the complex signaling pathways between the primary tumor and stromal cells, especially the immune cells [7,8].

Histotripsy is a novel, non-invasive, non-ionizing, and non-thermal ablation technique that mechanically destroys target tissue by controlled acoustic cavitation [9,10,11]. Using high-pressure (P– > 15 MPa), microsecond-length ultrasound pulses, at a low duty cycle (ultrasound on-time/total treatment time <1%), endogenous nanometer-scale gas nuclei in the tissue are expanded to over 100 µm, followed by collapse within several hundred microseconds, generating high mechanical stress and strain to disrupt the cells in the target tissue. Histotripsy has been shown to completely ablate the target tissue into a liquid consistency acellular homogenate, which is resorbed by the body within 2 months, leaving mm-length scar tissue [12,13,14]. Pre-clinical histotripsy investigations have established the feasibility and efficacy of histotripsy for non-invasive ablation, in many pre-clinical applications, including the ablation of human-scale porcine livers [15,16,17,18,19], and ablation of liver [20,21,22], kidney [23] and prostate [24] tumors. The mechanical strength of different tissue types impacts their resistance to histotripsy-induced damage, which can be exploited by selecting optimal parameters for achieving tumor-selective damage, while simultaneously protecting the structural integrity of large blood vessels, nerves, and bile ducts, within the ablation zone [15,17,18,25].

There has been some evidence suggesting the potent immunostimulatory and systemic effects of histotripsy. In a murine melanoma model, histotripsy ablation of the primary subcutaneous tumor inhibited the development of secondary pulmonary metastases, derived from tail vein injection [20]. In an orthotopic rat liver tumor model, histotripsy of the entire tumor volume resulted in complete regression of the tumor, in all subjects, and even in 5/6 cases of partial tumor ablation (50–75% tumor volume ablated), the entire tumor regressed completely, with no recurrence [22]. Recently, a first-in-human trial demonstrated the feasibility of planned histotripsy ablation of liver tumors, with no identified serious adverse events [26]. In all patients (*n* = 8), targeted tumors were locally controlled and in 2 of 8 patients, non-targeted tumors also stabilized [26]. These results suggest that local histotripsy ablation of the entire primary tumor, or even a part of the tumor, may systemically influence untargeted tumors. There is insufficient evidence regarding the effects of histotripsy on the risk of recurrence and metastases following tumor debulking. Since histotripsy mechanically disrupts the tumor, it is possible that intact tumor cells may inadvertently detach from the primary tumor and disseminate into the lymphatic and/or circulatory system. Our understanding of the immunological response to histotripsy is also limited, especially in cases of partial histotripsy ablation, where untargeted tumor volume remains undamaged during treatment, but eventually regresses. This longitudinal study aims to evaluate the effect of partial histotripsy tumor ablation on untargeted local tumor progression, survival outcomes, risk of developing metastases and tumor immune infiltration, in an orthotopic, immunocompetent, metastatic rodent HCC model, for the first time.

## 2. Materials and Methods

### 2.1. Experimental Design

Orthotopic McA-RH7777 liver tumors were generated in immunocompetent Sprague-Dawley rodent hosts. Once the tumor measured a minimum of 5 mm in its largest dimension, animals were randomized into treatment and control groups. For the survival study, *n* = 11 treatment rats received partial histotripsy ablation (approximately 50–75% of the tumor volume was targeted for ablation by histotripsy) and *n* = 11 control rats received no treatment. Animals were monitored weekly using MRI for up to 3 months or until the animals reached end-stage illness criteria, or the maximum tumor size (25 mm in largest dimension) allowed by ethical standards was reached; therefore, data are not truly absolute for animal survival. At the endpoint (prior to the onset of death), the animals were euthanized, and liver tissue was harvested for histological evaluation. For the early timepoint study, animals were euthanized at day 2 (*n* = 2 control, and *n* = 3 treatment) or day 7 (*n* = 3 control, and *n* = 6 treatment) post-histotripsy timepoint and the harvested liver tissue was analyzed to evaluate for intra-hepatic metastasis and immune infiltration at day 2 and day 7 timepoints following histotripsy ablation. An additional cohort of *n* = 3 animals received scant histotripsy ablation (<25% tumor volume targeted by histotripsy) and were euthanized at day 7.

### 2.2. HCC Cell Culture

McA-RH7777 (ATCC^®^ CRL-1601™) cells were purchased from the ATCC cell line repository. The cells were cultivated in Dulbecco’s Modified Eagle’s Medium (DMEM) containing 4 mM L-glutamine, 4500 mg/L glucose, and 1500 mg/L sodium bicarbonate, supplemented with 10% FBS and 1 mL Gentamicin. The cells were maintained at 37 °C in a 5% CO_2_/95% humidified air atmosphere.

### 2.3. Animals

This study was approved by the Institutional Animal Care and Use Committee at the University of Michigan and all procedures were performed in compliance with the approved protocol. Sprague-Dawley rats weighing 125–175 g were purchased from Taconic (Hudson, New York, NY, USA) and housed and maintained in specific pathogen-free (SPF) conditions in University of Michigan ULAM (Unit for Laboratory Animal Medicine) housing facility.

### 2.4. Orthotopic Tumor Implantation

Tumor implantation was performed via laparotomy. Animals were injected with 100 mg/kg cyclophosphamide intraperitoneally 24 h prior to tumor inoculation. For the inoculation procedure, animals were anesthetized in an induction chamber using isoflurane inhalation (5%, admixed with 1 L/min of oxygen) until loss of withdrawal reflex. After induction, the animals were moved to the surgical area in a supine position and anesthesia was then maintained with 2% isoflurane in 100% oxygen with a flow of 1.5 L/min administered using a nasal-cone connected to a coaxial breathing circuit and vaporizer (SurgiVet V704001, Smiths Medical, Waukesha, WI, USA). To prevent anesthesia-related corneal damage, eye lubricant was used. Carprofen (Rimadyl, Pfizer, NY, USA) analgesic (5 mg/kg) was used for analgesia prior to surgery and once every 24 h for two days post-surgery. The surgical area (chest and abdomen) was shaved and sterilized using chlorhexidine and iodine. Once the animal was draped and prepped for surgery, a midline incision was made through the skin. Blunt dissection was performed to separate the skin and abdominal muscle layer posteriorly and anteriorly to the ends of the incision. To expose the liver, an incision was made in muscle layer and retractor was inserted to keep the incision open. The inferior liver lobe was retracted using forceps. 10 million McA-RH7777 cells were suspended in 100 µL total injection volume constituted of basement membrane matrix (Matrigel, Corning Life Sciences, Corning, NY, USA) and serum-free DMEM - Dulbecco’s Modified Eagle Medium (Thermofisher Scientific, Waltham, MA, USA) in a 1:1 ratio. The cells were injected into the liver lobe. Pressure was applied on the injection site using sterile cotton tip applicator and a sterile hemostatic compressed sponge was placed on the liver surface to prevent leakage. After 3 min, the sponge was removed, and the muscle layer was closed using absorbable sutures. The skin incision was closed using wound staples. Animals were recovered until ambulatory. The wound staples were removed prior to pre-treatment MRI.

### 2.5. Histotripsy Ablation

Our rodent histotripsy setup (Figure 1a) consisted of a custom-built 1 MHz therapy transducer, co-aligned with a 20 MHz B-mode ultrasound imaging probe (L40-8/12, Ultrasonix, Vancouver, Canada) mounted to a motorized 3-axis positioning system [22,27]. The ring configuration transducer (f number = 0.6, focal distance = 32.5 mm) contains 8 individually focused lead zirconate titanate elements (20 mm diameter). The value of peak negative pressure in the tissue is estimated based on pressure measurements from fiber optic hydrophone in free-field [28]. In the free-field medium (degassed water), the pressure measurements from individual transducer elements were summed to calculate the peak negative pressure (P–) of 37.8 MPa and the peak positive pressure (P+) of 43.9 MPa. The peak positive pressure may be underestimated without considering the non-linearity developed when all elements are fired. However, the acoustic waveform at that high pressure with all elements fired simultaneously could not be measured due to instantaneous cavitation generation. The spatial peak temporal peak intensity (I_SPTP_) was estimated to be ~130 kW/cm^2^, spatial peak pulse average intensity (I_SPPA_) was estimated to be ~24 kW/cm^2^ and spatial peak temporal average (I_SPTA_) was estimated to be ~8 W/cm^2^. The transducer and imaging probe were immersed in a tank of degassed water maintained at 35–37 °C. For histotripsy treatment, animals were anesthetized by inhalation of isoflurane gas (1.5–2.0%) in 1 L/min of oxygen (SurgiVet V704001, Smiths Medical). The chest and abdominal regions were shaved with an electric clipper. Carprofen (Rimadyl, Pfizer, NY, USA) analgesic (5 mg/kg) was used for analgesia prior to histotripsy and once every 24 h for two days post-ablation. The animal was placed on a custom-built platform over the tank such that the liver region was submerged in the degassed water, which is the coupling medium (Figure 1a). Under ultrasound guidance, the therapy transducer was positioned to co-localize the focal zone of the therapy transducer focus with the tumor core. The entire tumor volume was visualized by scanning the imaging probe mounted on the motorized positioning system to determine the limits of the intended target volume. In the scant histotripsy ablation cohort, <25% tumor volume was targeted for ablation. In all other animals receiving histotripsy, 50–75% tumor volume was targeted. The desired fraction of tumor volume was targeted for histotripsy ablation by mechanically scanning the histotripsy focus to cover the targeted volume by using the motorized positioning system with continuous scan motion (scan velocity ~6 mm/s). At each focal location within the target volume, the therapy transducer delivered 1–2 cycle length histotripsy pulses (with a single high amplitude negative pressure phase) at 100 Hz PRF and generated peak negative pressure (P–) exceeding 30 MPa. P– >30 MPa exceeds the intrinsic threshold of the soft-tissue target (typically P– > 26 MPa) to generate inertial cavitation; this is the mechanism of intrinsic threshold histotripsy [28]. During histotripsy, the generated cavitation appeared hyperechoic compared to the surrounding liver parenchyma on ultrasound imaging (Figure 1b). The size of the cavitation cloud was 1–3 mm. Post histotripsy-ablation, the targeted region appeared hypoechoic, indicative of tumor tissue disruption (Figure 1b). After treatment, animals were recovered until ambulatory. Total ablation time ranged from 3–5 min, at 100 Hz PRF equated to total 18,000–30,000 pulses delivered to the intended target volume (ranging from 30 mm^3^ to 100 mm^3^). Histotripsy parameters were determined based on previous in vivo and in vitro work done in our lab [22,29].

### 2.6. MR Imaging

To assess tumor development, MRI was obtained within 1-day prior to histotripsy (pre-treatment timepoint) and within 1-day post-histotripsy (post-treatment timepoint). Weekly MRI was also used for monitoring ablation response. A 7.0 T MR small animal scanner using a Direct Drive console (Agilent Technologies, Santa Clara, CA, USA) was used with a 60 mm inner-diameter transmit–receive radiofrequency (RF) volume coil (Morris Instruments, Ottawa, ON, Canada). During imaging, rats were anesthetized using isoflurane inhalation (1.5–2.0% in 1 L/min of oxygen) and temperature was monitored using rectal probes. Respiratory gating was used during image acquisition. A custom-built proportional-integral-derivative (PID) controller (LabVIEW, National Instruments, Austin, TX, USA) was interfaced with a commercially available small animal system (SA Instruments, Stony Brook, NY, USA) to monitor respiration. Animal position in the scanner was confirmed with pilot scans. To visualize the tumor, a 2D T2-weighted fast spin-echo (FSE) in the coronal plane was used with the following parameters: (TR/TEeff = 2500/10 ms, FOV = 60 mm × 60 mm, slice thickness = 1 mm, data matrix (zero-filled) = 256 × 256 (512 × 512), resulting in a voxel size of 117 μm × 117 μm × 1000 μm, up to 28 slices acquired and total scan time ~5 min).

### 2.7. Histology

After euthanasia, treated tumor, as well as liver tissue samples were harvested and fixed in 10% buffered formalin for histopathological analysis. Fixed tissue samples were submitted to ULAM-IVAC (Unit for Laboratory Animal Medicine—In Vivo Animal Core, Ann Arbor, MI, USA) for paraffin embedding. Paraffin block samples were submitted to McClinchey Histology Labs, Inc. (Stockbridge, MI, USA) for sectioning in 4-micron thick slices and preparing unstained slides, Masson’s trichrome stained slides, and hematoxylin and eosin (H&E) slides. Slides were examined under high resolution microscope (KEYENCE BZ-X800, Keyence, Itasca, IL, USA) which was used to capture images.

### 2.8. Immunofluorescence Staining

For multicolor immunofluorescence antigen retrieval, the tissue sections were deparaffinized by passing 2 times in Xylene. The samples were then washed in Xylene Ethanol followed by pure ethanol and were then gradually rehydrated by passing through 70%, 50%, 30% ethanol, and distilled water for 3 min each. Masked epitopes from the sections were recovered by heat-induced antigen retrieval buffer by using water bath at 90 °C for 30 min. Two methods of heat-induced antigen retrieval were performed. Citrate buffer pH 6 (Abcam, Waltham, MA, USA) and/or Tris-EDTA pH 9 (Abcam) were used depending on the nature of the antibody. For CD11b, CD8 and NK1.1 staining, after antigen retrieval, sections were transferred to PBS (Phosphate Buffered Saline, Thermofisher Scientific) for 30 min, and then the sections were blocked in 5% BSA (Bovine Serum Albumin). After washing with PBS, the samples were incubated with a rabbit IgG specific for CD8a for 1 h at 37 °C. For visualization of CD8a, the sections were incubated in Alexa 555 labeled Goat anti-Rabbit IgG for 1 h at 37 °C. For visualization of CD11b and NK1.1, the sections were incubated with 1:100 diluted anti-CD11B antibody (clone M/170, Biolegend, San Diego, CA, USA) directly conjugated with Alexa 488 and anti-NK1.1 antibody (Biolegend) directly conjugated with Alexa 647 overnight at 4 °C.

For E-cadherin, N-cadherin, and vimentin staining, the sections were first incubated in 1:100 dilution of anti-vimentin mouse IgG, clone V9 (MilliporeSigma, Burlington, MA, USA) at 37 °C, and secondary antibody staining was performed using Alexa 488 goat anti-mouse IgG. After each secondary antibody incubation, samples were washed for 30 min in PBS to eliminate cross reactivity for next cycle of antibody staining. Sections were then incubated with 1:100 anti-N-cadherin antibody, clone 6A9.2 (MilliporeSigma) for 1 h at 37 °C. Secondary antibody staining was carried out using 1:100 Alexa 555 goat anti-mouse for visualization. After PBS wash, the sections were incubated with anti-E cadherin antibody [4A2] (Abcam) for 1 h at 37 °C. After washing with PBS, the sections were counterstained with Alexa 647 goat anti-mouse IgG, for visualization. After the final step of washing, the sections were quenched by using tissue autofluorescence quenching kit (Vector Biolabs, Malvern, PA, USA) and mounted using mounting media containing 4′,6-diamidino-2-phenylindole (DAPI, 1:10,000; MilliporeSigma).

### 2.9. Statistical Analysis

Statistical analysis was performed using SAS software (version 9.4, SAS Institute Inc., Cary, NC, USA). The difference between treatment and controls was assessed using one-way ANOVA. For survival data, Kaplan–Meier curves were generated. The survival time (defined as the time taken for animals to reach the tumor endpoint criteria, i.e., tumor burden greater than 25 mm in any single dimension) was compared between control and treated rats using the log-rank test. *p* < 0.05 was considered significant.

## 3. Results

### 3.1. Tumor Response to Histotripsy and Survival Outcomes

All animals tolerated the orthotopic liver tumor inoculation procedure without complication. Seven to nine days after inoculation, MRI imaging indicated a minimum tumor diameter of 5 mm in all animals. Histotripsy was performed successfully in all treatment cohort animals with no complications, and the post-procedure monitoring revealed no clinical issues.

In the survival cohort, 81.8% (*n* = 9/11) treatment animals experienced tumor burden reduction, following partial histotripsy ablation, and had tumor-free survival for the remainder of the study. In comparison, 100% (*n* = 11/11) untreated control animals demonstrated increased tumor burden and intrahepatic metastasis, and had to be euthanized within 1–3 weeks post-treatment timepoint (Appendix A). Histotripsy-treated animals had statistically significant improved survival outcomes compared to controls, with a *p*-value < 0.0001 (Figure 2).

The survival time in the control group was 1.45 ± 0.69 (mean ± SEM) weeks. In the treatment group, the survival time was 10 ± 0.84 weeks. All survival times are reported post-histotripsy timepoint, which was two weeks post-tumor inoculation. The survival time range was 1–3 weeks (control group) and 6–12 weeks (treatment group) after histotripsy timepoint. *n* = 7/11 animals in partial ablation group A were alive at 12 weeks, with no observable tumor, and were euthanized due to study endpoint, per our protocol. In these animals, even with partial ablation of the tumor, we observed complete regression of both ablated and untargeted tumor. *n* = 2/11 animal in the treatment group were euthanized at 6 weeks due to tumor burden greater than 25 mm in any one dimension. As such, 2/11 histotripsy animals with complete regression and no metastases were euthanized early at 7 weeks, due to research shutdown during COVID.

### 3.2. Radiology Observations

At the histotripsy timepoint (7–9 days post tumor inoculation), untreated tumors appeared hyperintense on T2-weighted MRI, compared to the adjacent normal liver tissue (Figure 3a). The tumor volume was 86.25 ± 15.12 mm^3^ (mean ± SEM) in the histotripsy group and 123.20 ± 25.87 mm^3^ (mean ± SEM) in the control group. The difference was not statistically significant (*p* = 0.232). Subsequent timepoints are measured from the histotripsy timepoint (week 0).

In all control animals, a multinodular primary tumor, along with multiple secondary metastatic nodules, was observed in 1–3 weeks (Figure 3a), and the animals had to be euthanized due to increased tumor burden.

In the treatment group, the ablated tumor region demonstrated a hyperintense appearance (if the imaging was performed >6 h after treatment) (Figure 3b) or demonstrated a hypointense appearance (if the imaging was performed within 4 h of treatment) (Figure 3c). The untargeted tumor region showed a similar appearance to the control tumor. In 9/11 animals, both ablated and unablated tumors began to demonstrate regression at 1 week post-treatment, and there was no detectable tumor on the MRI by 3 weeks (Figure 3b). In these animals, no recurrence or metastasis was observed until the study endpoint was reached. Of note, local tumor progression and metastases were observed in 2/11 animals, leading to increased tumor burden and the animals were euthanized at 6 weeks post-treatment (Figure 3c). No off-target ablation damage to skin or surrounding organs in any histotripsy animals was observed visually or on MRI images acquired post histotripsy.

### 3.3. Histology Observations in Survival Groups

In control animals, multinodular local tumor progression was observed on H&E staining (Figure 4a). In these cases, aggressive tumor growth occupied most of the liver lobe, both at the site of the original tumor and Intrahepatic metastases (Figure 4a). In treated animals demonstrating complete tumor regression, there was ~1 mm residual scar tissue, with scattered dystrophic calcification at the site of the original tumor, and no evidence of viable tumor cells on H&E staining (Figure 4b). These histology observations correlate with the MRI observation of no detectable tumor at the original treatment site, indicating complete regression of these tumors, with resultant formation of focal scar tissue. In treated animals demonstrating tumor progression, collagenous tissue with regions of dystrophic calcification was observed in the ablation cavity (Figure 4c). The local tumor progression of the residual, untargeted tumor had a similar appearance to the control tumor.

### 3.4. Histology Observations in Early Timepoint Groups

Trichrome staining of the control tumor on day 2 revealed nodular tumor extensions from the primary nodule, as well as areas of collagen deposition in the tumor core, indicative of necrotic regions (Figure 5a). In comparison, at day 2 post-histotripsy, there is a thin rim of immune cells surrounding the periphery of the ablated tumor and liver (Figure 5b). There are intact tumor cells, adjacent to the ablation zone, indicative of the untargeted, residual tumor. The core of the ablation zone shows blood products.

The control tumor at day 7 shows invasive appearance at the tumor periphery and multiple secondary tumor nodules (Figure 5c). In comparison, at day 7 post-histotripsy, there is evidence of homogenate resorption, with fibrotic or scar tissue beginning to form (Figure 5d). Inflammatory cells are scattered within the scar tissue and there is no evidence of intact tumor cells. On day 7 after scant histotripsy, collagenous deposition is observed, but there is no evidence of dystrophic calcification (Figure 5e). While there is evidence of blood products at the core, inflammatory cells are mainly confined to the ablation zone periphery. The appearance of the ablation zone is similar to the day 2 appearance post-histotripsy. These results suggest that the process of homogenate resorption and formation of scar tissue may be protracted in the scant histotripsy cohort, as compared to the histotripsy cohort.

### 3.5. Observations from Immunoflourescence Staining of Immune Cells

To determine the immune effects of histotripsy, infiltration of CD11b^+^ cells, CD8^+^ T cells and NK cells was compared in histotripsy-treated tumors vs. control tumors. At the core and the periphery of the control tumor, there is minimal immune infiltration on day 2 (Figure 6a). On day 2, increased immune infiltration of CD11b^+^ and NK cells is observed at the ablation zone periphery, compared to the control tumor periphery (Figure 6b). NK cells are also detected in the ablation zone on day 2 (Figure 6b).

On day 7, there is some infiltration of CD8^+^ T cells at the core and periphery of the control tumor, as compared to day 2 (Figure 6c). CD8^+^ T-cell infiltration is also observed at the boundaries of the histotripsy ablation zone, while NK cells and CD8^+^ cells are detected in the ablation zone core on day 7 (Figure 6d). In case of scant histotripsy ablation, reduced immune infiltration of CD8^+^ and NK cells is observed at the boundary of untreated tumor and normal liver in comparison to day 2 and day 7 post-histotripsy (Figure 6e). Similarly, within the scant histotripsy ablation zone, the immune infiltration is insubstantial (Figure 6e). Overall, increased immune infiltration is observed in histotripsy tumors, compared to controls on day 2 and day 7 post-treatment timepoint. It is possible that there is a minimum tumor fraction threshold that should be ablated to generate meaningful immune effects, as evidenced by the diminished immune infiltration observed in scant histotripsy (<25% volume ablation) vs. histotripsy treated tumors (50–75% volume ablation).

### 3.6. Observations for Immunoflourescence Staining for Epithelial and Mesenchymal Markers

To explore the impact of histotripsy on the risk of metastases, immunohistochemical expression of mesenchymal markers, i.e., loss of E-cadherin and gain of N-cadherin and vimentin, was compared in histotripsy-treated tumors vs. control tumors. The plasma membranes of hepatocytes express N-cadherin (Figure 7). There is weak expression of vimentin in the untreated control tumor (Figure 7a). At day 2 post-ablation, vimentin is upregulated at the ablation zone periphery (Figure 7b). At day 7, vimentin is also upregulated at the periphery of the control tumor (Figure 7c). At day 7 post-ablation, vimentin upregulation appears to have shifted inwards, towards the core of the ablation zone, while E-cadherin expression is upregulated at the periphery (Figure 7d). At day 7 post scant histotripsy, vimentin is weakly expressed at the perimeter of the untreated tumor, as compared to the control tumor at day 7 (Figure 7e). Vimentin expression is also up-regulated near the edge of the ablation zone, at the ablation zone-untreated tumor boundary on day 7 post scant histotripsy (Figure 7f).

The upregulation of the mesenchymal marker vimentin could be associated with either tissue healing or metastasis. Since there was evidence of scar tissue formation but no intact tumor cells on histology at day 7 post-histotripsy, the expression of vimentin at the periphery of the ablation zone on day 2, and in the ablation zone on day 7, suggests that tissue healing is the likely process. The similarity between the boundaries of the ablation zones on day 7 post scant histotripsy and day 2 post-histotripsy further supports the premise that the process of homogenate resorption and tissue healing may be protracted in scant histotripsy. Vimentin upregulation at the control tumor periphery and untreated tumor in scant histotripsy on day 7, may be linked to metastatic invasion. The invasive appearance of the control tumor on day 7 histology, as well as eventual development of metastases, in all control animals and *n* = 2 histotripsy animals with local tumor progression, support this proposition.

## 4. Discussion

Previous histotripsy investigations have demonstrated the feasibility and efficacy of histotripsy for non-invasive ablation in the liver [15,16,19,22,27]. In this study, we evaluated the development of metastatic HCC, after partial histotripsy tumor ablation, in an immune-competent, McA-RH7777 rodent liver tumor model. The McA-RH7777 rodent tumor model is a well-established rat orthotopic HCC model, used for image-guided interventional oncology research [30,31]. Even with partial ablation of the tumor volume, >80% histotripsy-treated animals demonstrated local tumor regression, with no local recurrence or metastasis, and had significantly improved survival, as compared to 100% control animals, demonstrating local tumor progression and intra-hepatic metastases. The formation of scar tissue was observed in histotripsy-treated animals, with no viable tumor cells observable, as early as day 7 post-histotripsy, indicating the resorption of the ablation zone begins early post-treatment. These results suggest that histotripsy does not increase the risk of developing metastases post-ablation. Increased immune infiltration was also observed in treated tumors at day 2 and day 7 post-histotripsy, as compared to control tumors, which may have triggered an anti-tumor immune response, contributing to the complete regression of partially ablated tumors, while preventing metastases in the immunocompetent rodent hosts.

Metastasis is a complex process, consisting primarily of five essential steps: (1) invasion, where tumor cells detach from ECM and infiltrate adjacent tissue, (2) intravasation, where tumor cells enter the circulatory system, (3) survival in the circulatory system, (4) extravasation, where tumor cells exit the circulatory system and infiltrate a distant site, and (5) colonization, where tumor cells grow and proliferate at the new site [32]. The seed and soil theory of metastasis was first proposed by Stephen Paget in 1889, which suggests that the metastatic growth of cancer cells (the ‘seed’) is dependent on the competence of the distant organ (the ‘soil’) [33]. HCC is a highly invasive cancer, favoring proximal intrahepatic metastasis, likely due to the dense hepatic vasculature and immunosuppressive polarization of the liver [34]. In this study, we focused investigation on the first step of the metastatic cascade, invasion, which is induced by epithelial-to-mesenchymal transition (EMT) of tumor cells, where they lose epithelial characteristics and concomitantly acquire mesenchymal characteristics. EMT is a biologic process that allows epithelial cells to transform into a mesenchymal cell phenotype, allowing them to migrate and infiltrate and can be classified into three biological subtypes. Of these, ‘type 2’ are associated with wound healing and organ fibrosis, and ‘type 3’ are associated with tumor progression and metastases [35]. EMT is characterized by the downregulation of epithelial markers, such as E-cadherin, and the upregulation of mesenchymal markers, such as N-cadherin and vimentin. In type 3 EMT, carcinoma cells, at the invasive front of primary tumors, can acquire a mesenchymal phenotype [36]. In our study, vimentin upregulation, observed at the periphery of control tumor on day 7, but not on day 2, is likely indicative of a type 3 EMT process. This premise is supported by the more invasive appearance of the tumor observed on day 7 trichrome staining, as well as the eventual invasive metastatic progression observed in all control tumors at later timepoints. In contrast, type 2 EMT is initiated as a reparative-associated process, in response to tissue injury or inflammation, and ceases once the inflammatory offense is withdrawn, such as with wound healing and fibrosis [37]. However, within the liver, the contribution of type 2 EMT to fibrosis is still controversial [38,39,40]. In general, vimentin is upregulated in the wound healing process; cytoskeletal vimentin is released extracellularly after tissue injury and binds to mesenchymal leader cells located at the wound edge to facilitate healing [41].

In our study, vimentin was observed at the histotripsy-treated tumor boundary on day 2 and within the tumor on day 7. This evidence suggests that the expression of vimentin is likely linked to the formation of fibrous tissue, as part of the wound healing process, in response to histotripsy ablation. Upregulation of E-cadherin, observed at the tumor boundary on day 7, is likely indicative of re-epithelialization of the ‘wound’, caused by histotripsy ablation; re-epithelialization is a necessary and essential requirement for successful wound closure [42]. Since EMT may be involved in both tissue healing and metastasis processes, further investigation of the metastatic cascade is necessary. Intravasation of tumor cells generates circulating tumor cells (CTCs); however, it has been estimated that only <0.01% of CTCs will develop distant metastatic lesions after surviving stress, immune attack and anoikis in a hostile circulatory system environment [43]. Additional studies are ongoing to assess the circulating tumor cells (CTCs), which will provide an insight into whether the mechanical disruption caused by histotripsy increases CTCs, as compared to untreated controls.

The immune microenvironment also affects the metastatic potential of the disseminating cells. Cancer therapies, such as radiation, chemotherapy and thermal ablation, destroy tumors and surrounding tissues via necrosis, which can trigger an inflammatory immune response. However, this response can be either pro-tumor or anti-tumor, depending on the tumor microenvironment and the expression of immune mediators and modulators [44,45,46,47]. Our previous study demonstrated that subcutaneous histotripsy ablation of melanoma tumors in murine hosts releases tumor antigens, with preserved immunogenicity, initiating both local (upregulation of intratumoral NK cells, dendritic cells, neutrophils, B cells, CD4^+^ T cells and CD8^+^ T cells), and systemic immune response, as evidenced by abscopal immune effects [20]. In the current study, we observed increased immune infiltration of CD11b^+^ and NK cells in the histotripsy-treated orthotopic liver tumor boundary, compared to control tumor at day 2 timepoint. At the day 7 timepoint, CD8^+^ T cells were also seen infiltrating the tumor region from the periphery. CD11b^+^ myeloid cells, as well as NK cells, contribute to anti-tumor innate immunity, while CD8^+^ cells contribute to adaptive immune-mediated responses [48,49,50]. Studies have shown that enhanced T-cell response, generated by immunotherapy, can prevent metastasis in early-stage cancer patients [51]. In a human colorectal cancer study, the absence of metastasis correlated with increased expression of T-cell proliferation and antigen presentation functions [52]. There is increasing evidence that ablative therapies can either positively or negatively impact tumor progression and metastasis, by regulating both adaptive and innate immunity [53,54,55,56]. A murine study reported that RFA of the liver induced a strong, time-dependent immune response (presence of neutrophils, activated myofibroblasts, and macrophages) at the necrotic zone boundary [57]. In a study involving colorectal cancer patients, incomplete RFA (presence of remnant tumor after therapy) was shown to promote tumor progression and was associated with earlier development of metastases [58]. Another study reported increased metastatic potential of residual carcinoma after transarterial embolization, using the McA-RH7777 model [31]. The first in-human histotripsy study reported intrahepatic abscopal effects (reduction in non-targeted tumor lesions), following histotripsy ablation of a single liver tumor lesion in a colorectal cancer patient, with progressive, multiple metastatic disease [26]. Our previous study, using the N1-S1 liver orthotopic tumor model, had demonstrated that the complete histotripsy ablation cohort had a 100% regression rate; however, in the partial ablation cohort (50–75% tumor ablation), only 80% of animals demonstrated regression [22]. Similar results were observed in the current study; when at least 50% tumor volume was ablated, >80% of the tumors completely regressed and no clinical recurrence/metastases were detected, indicating an anti-tumor immune response. However, in the scant histotripsy cohort (<25% tumor ablation), tumor progression was observed, suggesting that the anti-tumor immune response was too weak, as evidenced by the diminished immune infiltration. There is likely a minimal tumor volume percentage threshold that needs to be ablated to generate sufficient histotripsy response to cause the entire tumor (both ablated and unablated regions) to regress.

One of the primary limitations of our study is that only a small subset of immune cells was analyzed. Future studies will focus extensively on the quantitation of additional intratumoral and peripheral immune cell subsets, using flow cytometry assays. Tumor progression and metastasis are influenced by several molecular mechanisms and factors. The orthotopic implantation of McA-RH7777 cells, using cyclophosphamide for temporary immune suppression to promote tumor uptake [59], does not fully represent spontaneously developing tumors in the human liver. This study also did not utilize any immunomodulating drugs in combination with histotripsy. Although histotripsy alone resulted in complete regression of the tumor in >80% animals, it would be worthwhile to explore the response of histotripsy in combination with immunotherapies, since ablation-induced immune responses have been historically reported as inadequate in eliminating established tumors. The exact mechanism of cell death of the untargeted residual tumor, following partial ablation, is also unknown and is likely a combination of immunogenic cell death mechanisms; further investigations are currently ongoing. Another limitation is that tumors in this study were treated at an early stage, before metastases were radiologically visible. Future studies will utilize immunomodulatory adjuvants with histotripsy ablation to treat tumors in different stages of disease progression, including after the development of intrahepatic metastases, and monitor their response.

## 5. Conclusions

This study demonstrated the potential of histotripsy for successful non-invasive tumor ablation, and prevention of local tumor progression and metastasis. Even with partial ablation, complete local tumor regression was observed in 9/11 (81%) treatment rats, with no recurrence or metastasis up to the 12-week study endpoint, as evidenced by MRI and histology. In contrast, 11/11 (100%) control animals demonstrated local tumor progression and intra-hepatic metastases and had to be euthanized at 1–3 weeks. The results also revealed increased immune infiltration in histotripsy-treated tumors, as compared to controls as preliminary evidence, which suggests a possible immune-mediated response, stimulated by histotripsy, which may have contributed to regression of the untreated tumor region. These results suggest that histotripsy may not increase the risk of developing metastases post-ablation, as compared to controls. Future studies will continue to investigate the safety, efficacy, and biological effects of histotripsy, for potential translation to clinic.

## Figures and Tables

**Figure 1 cancers-14-01612-f001:**
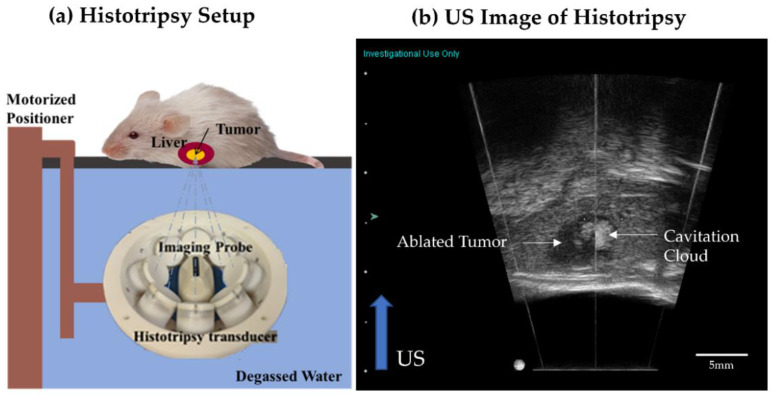
(**a**) The rodent histotripsy treatment setup consisted of an 8 element 1 MHz therapy transducer delivering 1–2 cycle pulses at P– 30 MPa and 100 Hz PRF. A coaxially aligned 20 MHz imaging probe was used for real-time ultrasound guidance. Both transducers were mounted to a motorized positioning system and immersed in a tank of degassed water (coupling medium). The animal was placed in a prone position on a platform to allow the intended target region to submerge. (**b**) Generation of hyperechoic cavitation cloud in liver tumor. The ablated tumor region appears hypoechoic compared to surrounding liver parenchyma.

**Figure 2 cancers-14-01612-f002:**
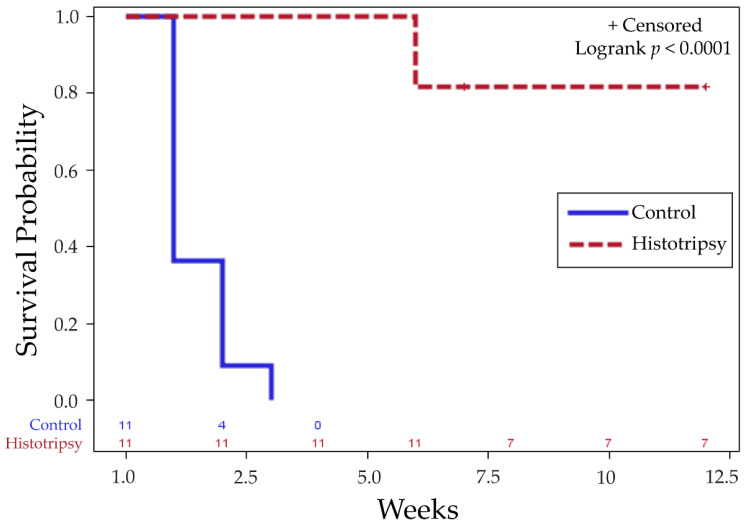
Kaplan–Meier survival curve indicates significant difference in survival outcomes of histotripsy-treated animals vs. untreated controls for tumor progression (*p* < 0.0001). ‘+ Censored’ indicates that observations are right censored as *n* = 2/11 histotripsy animals still surviving at 7 weeks with no observable tumor were euthanized due to COVID shutdown.

**Figure 3 cancers-14-01612-f003:**
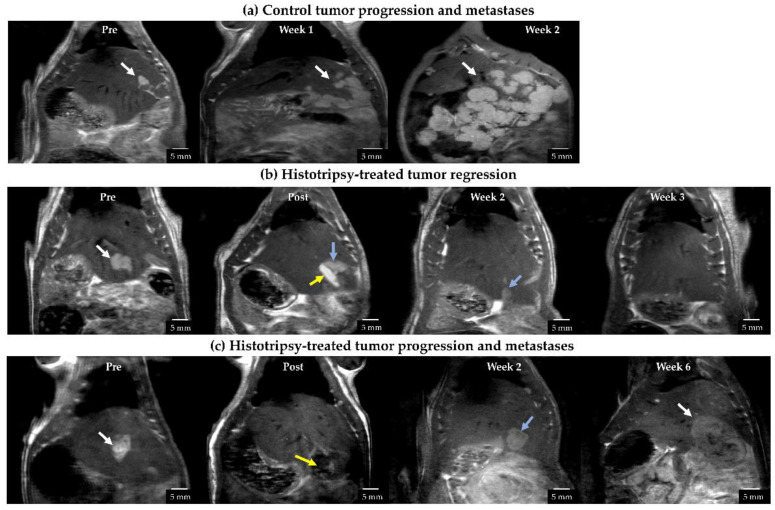
Representative T2-weighted MR images for (**a**) untreated control, (**b**) histotripsy-treated tumor showing complete regression, and (**c**) histotripsy-treated tumor with local tumor progression. (**a**) Untreated control tumor appeared hyperintense compared to surrounding liver parenchyma at pre-treatment timepoint, developed metastases at week 1 which grew aggressively by week 2. White arrows show the tumor location. (**b**) Post-ablation, the ablated region appears hyperintense (yellow arrow) compared to untargeted tumor (blue arrow) likely due to edema (image acquired 8 h post-treatment). By week 2, tumor appeared to regress (blue arrow) and was undetectable from week 3 onwards. (**c**) Post-ablation, the ablated region (yellow arrow) appears hypointense (image acquired within 2 h post-treatment). At week 2, the tumor did not show signs of size regression (blue arrow). Local tumor progression and metastases were observed by week 6 (white arrow).

**Figure 4 cancers-14-01612-f004:**
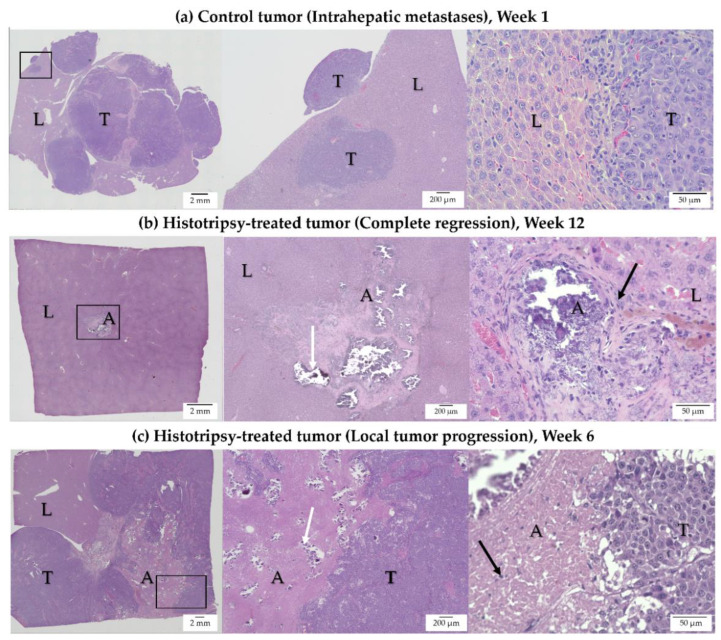
H&E-stained representative images for (**a**) untreated control, (**b**) histotripsy-treated tumor showing complete regression, and (**c**) histotripsy-treated tumor with local tumor progression (L: Liver, A: Ablation Zone, T: Tumor). For each row, the first panel shows a low-magnification view of the entire tissue section, with an identified region of interest (ROI), the second and third panel show the ROI at higher magnifications. (**a**) Control tumor—Week 1: Intra-hepatic tumor progression and metastases are observed in the untreated control. Tumor growth occupied most of the liver lobe. (**b**) Histotripsy-treated tumor—Week 12: In a treated tumor demonstrating regression, ~1 mm scar tissue with scattered dystrophic calcification in the ablation zone (white arrow) and inflammatory cells (black arrow) within and surrounding the ablation zone are observed. (**c**) Histotripsy-treated tumor—Week 6: In a treated tumor demonstrating local tumor progression, multinodular tumor growth is observed. Collagenous tissue with scattered dystrophic calcification is observed in the ablation zone (white arrow). A few inflammatory cells are seen in the ablation zone (black arrow).

**Figure 5 cancers-14-01612-f005:**
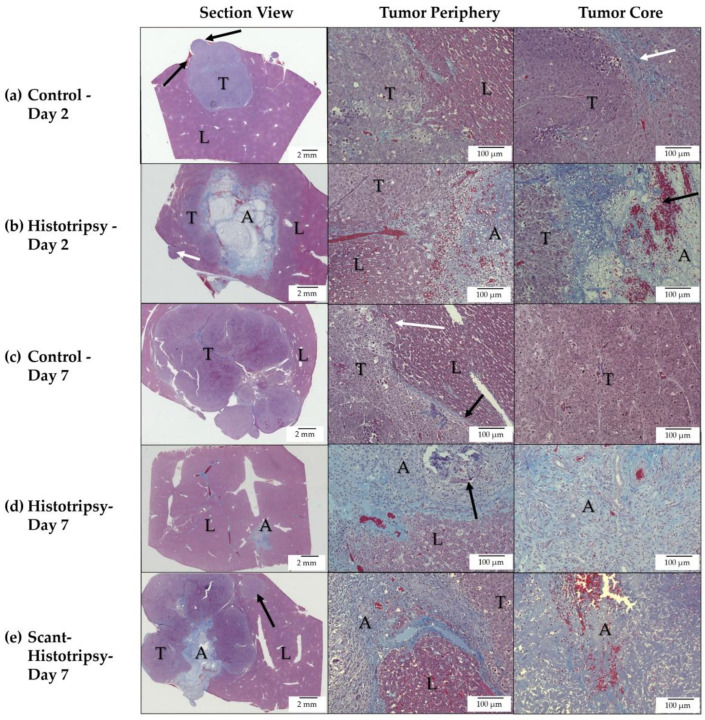
Trichrome-stained representative images for (**a**) untreated control tumor (Day 2), (**b**) histotripsy-treated tumor (Day 2), (**c**) untreated control tumor (Day 7), (**d**) histotripsy-treated tumor (Day 7), and (**e**) scant histotripsy-treated (<25% tumor volume ablated) tumor (Day 7). All days are measured from the histotripsy-treatment timepoint. (L: Liver, A: Ablation Zone, T: Tumor). (**a**) Control—Day 2: Nodular tumor extensions from the primary nodule are observed (black arrows), areas of collagen deposition are observed in the tumor core (white arrow). (**b**) Histotripsy—Day 2: The ablation zone can be distinguished from the untreated residual tumor region. A metastatic nodule is observed (white arrow). The ablation zone consists of mostly acellular debris with scattered red and white blood cells (black arrow). (**c**) Control—Day 7: Local tumor progression of the primary tumor and multiple metastatic nodules are observed, colonizing most of the liver lobe. Tumor cells at the periphery demonstrate invasive characteristics (white arrow), at the location where a thin rim of collagenous tissue separating the tumor from normal liver is breached (black arrow). (**d**) Histotripsy—Day 7: Tumor is replaced by scar tissue with areas of dystrophic calcification (black arrow), substantial infiltration of inflammatory cells, and few red blood cells. No viable tumor cells are observed. (**e**) Scant Histotripsy—Day 7: Local tumor progression of the primary tumor and a metastatic nodule (black arrow) are observed with an area of scar tissue within the ablation zone. The ablation zone contains red blood cells and collagen. Inflammatory cells are observed mainly at the periphery of the ablation zone, but not at the core of the ablation zone.

**Figure 6 cancers-14-01612-f006:**
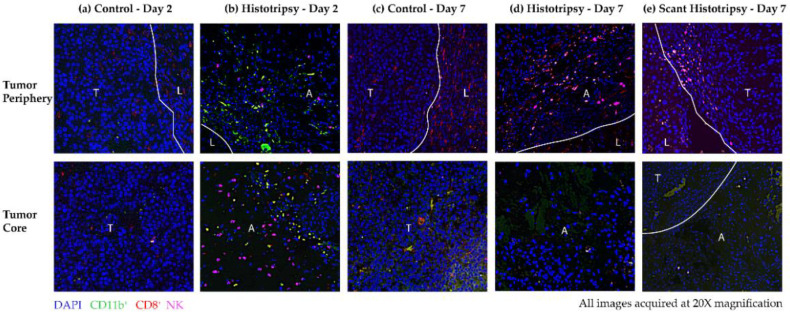
Representative multiplex immunohistochemistry images obtained at 20× magnification showing immune cell infiltration in (**a**) untreated control (Day 2), (**b**) histotripsy-treated tumor (Day 2), (**c**) untreated control (Day 7), (**d**) histotripsy-treated tumor (Day 7) and (**e**) scant histotripsy-treated (<25% tumor volume ablated) tumor (Day 7) at the tumor periphery (top) and tumor core (bottom). All days are measured from the histotripsy-treatment timepoint. (L: Liver, A: Ablation Zone, T: Tumor). The samples were stained for DAPI (blue), CD11b (green), CD8 (red) and NK (pink). (**a**) Control—Day 2: There is minimal immune infiltration at the core of the untreated control tumor. (**b**) Histotripsy—Day 2: At the periphery, increased infiltration of CD11b^+^ and NK cells is observed, NK cells are also observed at the core of the ablation zone. (**c**) Control—Day 7: There is some infiltration of CD8^+^ cells at the periphery, but no substantial infiltration is observed at the tumor core. (**d**) Histotripsy—Day 7: NK and CD8^+^ cells infiltrated the tumor periphery and were also detected at the core of the ablation zone. (**e**) Scant Histotripsy—Day 7: Some CD8^+^ and NK cells are observed at the untargeted tumor–liver interface, but no substantial infiltration is observed in the ablation zone.

**Figure 7 cancers-14-01612-f007:**
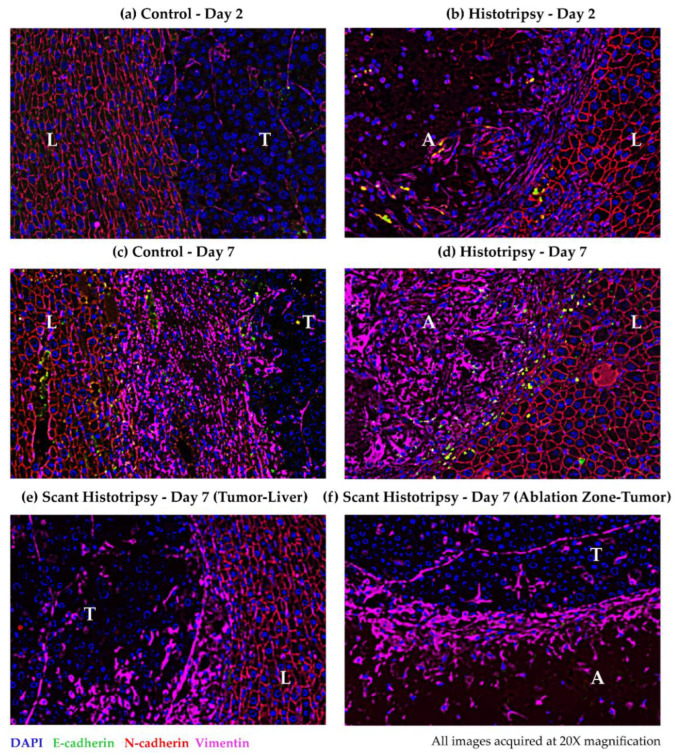
Representative multiplex immunohistochemistry images obtained at 20× magnification showing epithelial and mesenchymal markers in (**a**) untreated control (Day 2), (**b**) histotripsy-treated tumor (Day 2), (**c**) untreated control (Day 7), (**d**) histotripsy-treated tumor (Day 7), and (**e**,**f**) scant histotripsy treated (<25% tumor volume ablated) tumor (Day 7). All days are measured from the histotripsy-treatment timepoint. The samples were stained for DAPI (blue), E-cadherin (green), N-cadherin (red) and vimentin (pink) (L: Liver, A: Ablation Zone, T: Tumor). In all panels, N-cadherin is expressed at the plasma membrane of hepatocytes. (**a**) Control—Day 2: Vimentin is weakly expressed within the control tumor. (**b**) Histotripsy—Day 2: Vimentin expression is upregulated at the ablation zone periphery. (**c**) Control—Day 7: Vimentin expression is upregulated at the control tumor periphery. (**d**) Histotripsy—Day 7: Vimentin is up-regulated within the ablation zone, E-cadherin is expressed at the ablation zone periphery. (**e**) Scant Histotripsy—Day 7: Vimentin is expressed at the untreated tumor–liver interface. (**f**) Scant Histotripsy—Day 7: Vimentin is expressed in the periphery of the ablation zone at the ablation zone-untreated tumor interface.

## Data Availability

All relevant data are within the manuscript. Raw image data will be available upon request.

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
