# Peer review of "Impact of Histotripsy on Development of Intrahepatic Metastases in a Rodent Liver Tumor Model"

_cancers, 2022, doi:10.3390/cancers14071612_

Round 1

Reviewer 1 Report

No comments.

The paper in the present form is ok.

Reviewer 2 Report

No further comments

This manuscript is a resubmission of an earlier submission. The following is a list of the peer review reports and author responses from that submission.

Round 1

Reviewer 1 Report

   The manuscript of Worlikar et al. describes the use of histotripsy to treat a hepatocellular tumour model grown in immune-competent rats. The authors find that histotripsy treatments result in the regression of tumours in the vast majority of treated animals. As a result, a significant survival advantage for the treated animals compared to control animals is shown. In addition, the authors use histological techniques to show that these effects might be associated with infiltration of the tumours by immune cells and regulation of EMT markers such as vimentin. 
The subject under investigation would be of interest to the readers of this journal, the manuscript is well written and results are presented. The strongest point of this manuscript is the dominant tumour-regression related pro-survival effects the authors see when they treat the animals with histotripsy. These results represent one of the strongest pro-survival data sets published to date in any tumour model. 
Sadly the authors have not proceeded to an in-depth analysis of the immune response in different lymphoid organs and the tumour using appropriate assays. For example, the use of multicolour flow cytometry would provide stronger data for the activation of different immune cell subsets. Also, the use of functional assays would show which immune cells are causing the survival phenotype seen after histotripsy treatments. The biological impact of the paper suffers as a result. Also, improvement in the survival of subjects after histotripsy and/or HIFU, as well as systemic activation of the immune system resulting in long-term anti-cancer effects have been shown previously in other published studies.
For these reasons, I regret that I cannot recommend this article for publication in Cancers.

It might be appropriate for another reviewer to also have a look at it.

Author Response

Thank you for the insightful feedback. We believe that the promising pro-survival data in our pre-clinical study will have a broad impact on future research in this field.

While we agree that the immune analysis data may be limited, we think this study makes a valuable contribution to the field because this is the first 3-month study to demonstrate that ‘partial’ histotripsy ablation of only 50-75% tumor volume in a highly aggressive, metastatic, orthotopic liver tumor model reduced local tumor progression, did not increase the risk of liver metastases and improved survival outcomes. The primary focus of this study was to investigate whether and how histotripsy impacts the risk of metastases.  Metastases were measured with MRI and histology. Limited immune analyses were performed in this study to provide an explanation for why partially ablating the tumor might be causing the entire tumor (both ablated and untargeted tumor volume) to regress. Specifically, these immune subsets were chosen since CD11b+ myeloid cells, NK cells play a major role in anti-tumor innate immunity, while CD8+ T cells contribute to adaptive anti-tumor immunity. Using microscopy-based analysis, we have observed that the tumor immune infiltration post histotripsy is tumor region-specific, i.e. high infiltration concentrated in the residual, untargeted tumor and the tumor periphery, while the ablated region consists mostly of extracellular matrix and non-viable tissue. Such observations would not be possible with global assays like flow cytometry. In future studies, we will focus on in-depth analysis of both local and systemic immune response at different timepoints post histotripsy. 

We appreciate the time and effort that you and the other reviewers dedicated to providing feedback on our manuscript and are grateful for your valuable suggestions for improving our paper. We have incorporated the suggestions made by the reviewers and the editor. All changes to the original manuscript are marked within the manuscript file named ‘Cancers_Revised_Submission’ and are recorded in the attached file 'Response_CoverLetter_Cancers' attached below.

Reviewer 2 Report

This manuscript discusses the first study to evaluate the impact of partial histotripsy ablation on immune 28 infiltration, survival outcomes, and metastasis development in an in vivo orthotopic, immuno- 29 competent rat HCC model (McA-RH7777).

This study evaluated the effect of partial histotripsy tumor ablation on un- targeted local tumor progression, survival outcomes, risk of developing metastases and tumor immune infiltration in an orthotopic, immunocompetent, metastatic rodent HCC model.

Line 56: high intensity focused ultrasound (HIFU))

> Please delete double bracket.

Line 183: peak negative pressure (p-) exceeding 30 MPa

> Please specify the measured value peak negative pressure, peak positive pressure and if it is possible ultrasound power.

Line 189: Total ablation time ranged from 3-5 minutes..

> PRF = 100Hz most probably excludes treated tissues increasing temperature, have you performed temperature measurements during the insonations? If yes please add the results.

Line 194: The rodent histotripsy treatment setup consisted of an 8 element 1MHz therapy transducer

> Please add transducer brand name and model

Line 313: Figure 3. Representative T2- weighted MR images

> Please add tumours dimensions

Author Response

We appreciate the time and effort that you and the other reviewers dedicated to providing feedback on our manuscript and are grateful for your valuable suggestions for improving our paper. We have incorporated the suggestions made by the reviewers and the editor. All changes to the original manuscript are marked within the manuscript file named ‘Cancers_Revised_Submission’ and are recorded in the attached file 'Response_CoverLetter_Cancers' attached below.

Reviewer 3 Report

My comments are only on the ultrasound and imaging components and are optional suggestions to improve the paper.

l174 - Would it be possible to provide a sample ultrasound image.

l183 - How was the peak negative pressure measure/estimated? Details of dosimetry required if possible. Also what is the size of the ablation region for one sho.

l190 - was the ablation step and repeat or on the fly, if the latter what was the approx scanning velocity.

l268 this section is mainly about survival, and the increase or decrease in tumour burden is covered later (Section 3.2) so comments in this sentence about tumor burden should be removed and/or moved to Section 3.2.

l305 What is the untreated tumour in the treatment group, this is confusing?

l310 When and how was the lack of off-target damage assessed (visually, imaging, post-mortem etc)

Author Response

We appreciate the time and effort that you and the other reviewers dedicated to providing feedback on our manuscript and are grateful for your valuable suggestions for improving our paper. We have incorporated the suggestions made by the reviewers and the editor. All changes to the original manuscript are marked within the manuscript file named ‘Cancers_Revised_Submission’. Please find attached the zip file with updated submission materials.